# Relationship between Magnesium Intake and Chronic Pain in U.S. Adults

**DOI:** 10.3390/nu12072104

**Published:** 2020-07-16

**Authors:** Emily K. Tarleton, Amanda G. Kennedy, Gail L. Rose, Benjamin Littenberg

**Affiliations:** 1Department of Environmental and Health Sciences, Northern Vermont University, Johnson, VT 05661, USA; 2Department of Medicine, Larner College of Medicine, University of Vermont, Burlington, VT 05405, USA; amanda.kennedy@uvmhealth.org (A.G.K.); benjamin.littenberg@uvm.edu (B.L.); 3Department of Psychiatry, Larner College of Medicine, University of Vermont, Burlington, VT 05405, USA; gail.rose@uvmhealth.org

**Keywords:** magnesium, chronic pain, NHANES

## Abstract

Chronic pain is a public health concern and additional treatment options are essential. Inadequate magnesium intake has been associated with chronic pain in some populations. We sought to examine the relationship between dietary magnesium intake and chronic pain in a large, representative cohort of U.S. adults (NHANES). Of the 13,434 eligible adults surveyed between 1999 and 2004, 14.5% reported chronic pain while 66% reported inadequate magnesium intake. The univariate analysis showed a protective effect of increased magnesium intake adjusted for body weight (odds ratio 0.92; 95%; CI 0.88, 0.95; *p* < 0.001). It remained so even after correcting for socioeconomic and clinical factors as well as total calorie intake (odds ratio 0.93; 95% CI 0.87, 0.99; *p* = 0.02). The association was stronger in females (odds ratio 0.91; 95% CI 0.85, 0.98; *p* = 0.01) than males (odds ratio 0.96; 95% CI 0.89, 1.04; *p* = 0.32). The potential protective effect of magnesium intake on chronic pain warrants further investigation.

## 1. Introduction

The U.S. is in the midst of an epidemic of chronic pain and an opioid abuse crisis. Chronic pain is associated with dependence on opioids, anxiety, depression, and reduced quality of life [1] and leads to $560 billion in medical costs, lost productivity, and disability programs each year [2]. Healthy People 2020 includes a development objective to “decrease the prevalence of adults having high-impact chronic pain” [3].

Managing chronic pain is difficult. Pharmacological treatment options such as opioids can result in some relief, but come with significant risk of dependence, cognitive impairment, respiratory suppression, and constipation [4] and are not recommended for long term use. Other medications, such as nonsteroidal anti-inflammatory drugs, are less effective and carry their own risks including bleeding, kidney damage, and high blood pressure [4].

Non-pharmacological, non-invasive approaches have an important role, particularly considering the serious risks and uncertain benefits of other options [5,6,7]. Psychological treatments such as cognitive behavioral therapy and mindfulness meditation, and other non-pharmacological modalities such as exercise, multidisciplinary rehabilitation, acupuncture, and massage have a very low risk of harm, do not incur physiological dependence, and in many cases, are as efficacious as medications for reducing pain and increasing functioning and quality of life, although evidence is mixed [8,9,10,11,12,13,14,15,16]. Nonetheless, a sizable number of patients have pain resistant to all these modalities and additional treatment options are needed [17].

Nutrition may play a role in chronic pain and its management. In particular, there is some evidence that magnesium supplementation results in improved pain management and that magnesium influences inflammation and inflammatory markers [18]. As the fourth most abundant mineral and second most abundant cation in the cell, magnesium is a cofactor for over 300 enzymes and is essential for both aerobic and anaerobic energy production, glycolysis, mitochondrial oxidative phosphorylation, and potassium and calcium regulation [19,20]. Adequate magnesium intake is essential for energy production, prevention of dysrhythmias, blood pressure regulation, avoidance of insulin resistance, and bone homeostasis [21,22]. Inadequate magnesium intake has been associated with diabetes, cardiovascular disease, depression, and chronic pain [22,23,24]. Despite its importance, a majority of U.S. adults do not consume adequate amounts of dietary magnesium [25].

The mechanisms responsible for magnesium’s role in chronic disease, including chronic pain, are unclear. However, human studies suggest that oral magnesium is causally related to pain reduction [26]. Because chronic pain is so difficult to treat and magnesium supplementation is such a low-cost intervention, it is worthwhile exploring the relationship between the two. In this study, we sought to test the existence of a relationship between dietary magnesium intake and chronic pain reports using a large, cross-sectional population-based data set from the U.S.

## 2. Materials and Methods

### 2.1. Data Source and Subjects

We used data from the U.S. Center for Disease Control and Prevention’s (CDC) National Health and Nutritional Examination Survey (NHANES). NHANES participants undergo extensive interviews and laboratory assessments including measures of dietary intake, dietary supplements, socioeconomic factors, clinical characteristics, and personal habits [27]. Because of its sampling structure, NHANES can be used to represent the sex-, age-, race- and ethnicity-adjusted non-institutionalized population of the U.S. [28,29]. While dietary magnesium intake is available in all datasets, only the 1999–2004 datasets include chronic pain data. Thus, our sample for analysis included all subjects from the 1999–2004 datasets who were at least 20 years old with complete data for the outcome (chronic pain) and predictor (magnesium intake).

### 2.2. Variables

The outcome variable was chronic pain, defined as pain present for at least 3 months during the past year, including the past month.

The main predictor variable was total magnesium intake in milligrams/day/kilogram of body weight calculated from 24-h dietary and supplement recall data, and weight measured by NHANES staff. Intake was used due to the unreliability of serum magnesium levels [30], and because it is directly modifiable and could serve as an intervention. Inadequate magnesium intake was defined with age- and sex-varying thresholds taken from the Estimated Average Intake (EAR) as intake <350 mg/day for men over age 30, <330 mg/day for younger men, <265 for women over age 30, and <255 for younger women [31].

Based on review of the literature and our clinical experience, we considered age, sex, race, ethnicity, household income, education, marital status, total daily energy intake, daily physical activity level, tobacco use, alcohol use, and chronic inflammatory disorders (kidney disease, atherosclerosis, rheumatoid arthritis, osteoarthritis, chronic obstructive pulmonary disease (COPD), and diabetes) as potential confounders of the relationship between chronic pain and magnesium intake. Body mass index was calculated using the weight measured by NHANES staff in kilograms divided by height in meters squared. Household income was dichotomized as low if it was reported to be $35,000 per year or less. Education was dichotomized as having a high school diploma (or equivalent) vs. not. Marital status was classified as married or living as a couple vs. never married, single, divorced or widowed. Daily physical activity was reported in four distinct categories: sitting most of the day, walking around but no lifting or carrying, lifting light loads and climbing stairs or hills, or heavy work and carrying heavy loads. Tobacco use was considered present if the patient endorsed current smoking vs. absent for former smokers and those who never smoked. Alcohol use was coded as the average number of units consumed per week over the past year. A unit of alcohol is one can of beer, one glass of wine, or one ounce of liquor. Only adults aged 20 and over are asked questions about alcohol, hence the inclusion criteria of those 20 years and older. Chronic inflammatory disorders were considered present if the respondent endorsed that a doctor or other health professional had told them they had the diagnosis, or, in the case of diabetes and kidney dysfunction, if the diagnosis was indicated by laboratory studies.

### 2.3. Statistical Analysis

The primary hypothesis was that chronic pain is associated with magnesium intake per day per kilogram of body weight while adjusting for possible confounders. We used logistic regression to assess the relationship and adjust for potential confounders. Because magnesium intake varies with both age and gender [32], we constructed additional models to explore the possibility of interactions of magnesium with gender and magnesium with age. In secondary analyses, we considered total magnesium intake per day regardless of weight and total magnesium intake per day per kilocalorie of energy intake.

All analyses employed the stratification and weighting scheme recommended for NHANES by the National Center for Health Statistics [28] using Stata 15.1 (Stata Corp, College Station, TX, USA). *p* values less than 0.05 were considered to be statistically significant.

## 3. Results

Of the adult subjects in the 1999–2004 NHANES data set, 32% met eligibility criteria, for a final sample size of 13,435 (Figure 1). Of these, 14.5% reported chronic pain while 66% reported inadequate magnesium intake. Adults with chronic pain are significantly different from those without pain. Chronic pain sufferers consumed less magnesium and are more likely to be older, white, female, a current smoker, have low household income, be more physically inactive, and to report having an inflammatory disorder. See Table 1.

Adults with chronic pain had lower magnesium intake per kg of body weight than adults without chronic pain (control group) (−0.34 mg/kg/day; 95% confidence interval (CI) −0.47, −0.21; *p* < 0.001). The odds ratio for having chronic pain is 0.92 per mg/kg/day (CI 0.88, 0.95; *p* < 0.001), suggesting an 8% reduction in the odds for each mg of intake per kg per day. When adjusting for all potential confounders, the odds ratio is 0.93 (CI 0.87, 0.99; *p* = 0.02). See Table 2. This relationship varies by gender. Women demonstrated a strong protective association of magnesium intake on pain (odds ratio 0.91; 95% CI 0.85, 0.98; *p* = 0.01). In men, the relationship was much weaker (odds ratio 0.96; 95% CI 0.89, 1.04; *p* = 0.32). The relationship was stronger in younger adults (odds ratio 0.93; 95% CI 0.86, 1.00; *p* = 0.04) than in those over age 65 (odds ratio 1.00; 95% CI 0.92, 1.09; *p* = 0.99) (Figure 2).

In secondary analyses, the relationship of chronic pain to total magnesium intake per day (adjusted for all potential confounders and body weight), was null (odds ratio 1.000; CI 0.999, 1.001; *p* = 0.43). Likewise, the relationship of magnesium intake per energy intake was not significant (odds ratio 0.49; CI 0.07, 3.52; *p* = 0.47).

## 4. Discussion

Using data from the 1999–2004 NHANES, we confirmed that a majority of the U.S. adult population does not consume adequate amounts of magnesium. We also found that for every additional milligram of magnesium consumed per kilogram of body weight, the odds of experiencing chronic pain decreased by 7%. This finding is supported by previous research [31] and suggests that magnesium intake needs to be conceptualized (and perhaps managed) quite differently in adults of different sizes. Adults with chronic pain also differ from pain-free adults by socioeconomic factors. The relationship of magnesium intake to chronic pain is stronger in women and adults less than age 65. According to national data [1], and the results shown here, prevalence of chronic pain is higher in women. Women also tend to consume less dietary magnesium [33,34]. It is unclear whether a larger sample size would show a significant association in males.

These data show adults with chronic pain also differ from adults in the control group by the prevalence of inflammatory diseases. Although not definitive, they are consistent with a model of chronic pain as an inflammatory process caused by an underlying disorder. The mechanism by which magnesium intake is associated with chronic conditions, including inflammation and chronic pain, is not clear. However, magnesium’s role as a calcium antagonist may be important. Prolonged opening of calcium channels and activation of N-methyl-D-aspartate (NMDA) receptors in the absence of magnesium leads to a cascade of events, which favor the inflammatory process [35]. In the presence of increased concentrations of intracellular calcium, phagocytic cells are activated leading to increased cytokine production [36].

NMDA receptor antagonists, including magnesium, are becoming more widely studied as possible pharmacological pain management targets due to their role in pathophysiology of several neurologic and psychiatric disorders [37,38]. The use of magnesium for the treatment of chronic pain has been studied in patients with migraine headaches, complex regional pain syndrome, neuropathic pain, chronic low back pain, and fibromyalgia [26,37]. Studies reporting on the use of magnesium for the treatment of migraine headaches most often used intravenous magnesium sulfate [39,40] with mixed results, while studies in patients with fibromyalgia have used magnesium citrate tablets [41] or transdermal magnesium chloride [42] with positive results. One study of patients with chronic back pain combined 2 weeks of intravenous magnesium sulfate with 4 weeks of magnesium oxide tablets with positive results [43]. In general, intravenous magnesium is not a convenient or economical mode of administration. Better clinical trials are needed due to the limitations of these studies such as small sample sizes, varying doses and forms of magnesium, and short follow-up periods.

Counseling patients on foods containing magnesium, such as nut, seeds, and green leafy vegetables, is appropriate for chronic pain patients given the low percentage of adults meeting the daily requirements. While dietary changes take time, patients seeking to avoid addiction or side effects from chronic pain medications may find this option attractive. Magnesium supplements, if found to be efficacious, would combat pain more quickly than dietary changes alone. Supplements are low risk, tend to be low cost, and cause relatively few side effects [44]. Oral supplementation is safe in adults when used in dosages below 350 mg of elemental magnesium per day [31]. Nutrient supplements in general are widely accepted and available [45] and offer an opportunity as an alternative treatment.

This study has several strengths including the large sample size recruited from a nationally representative population. We were able to show the association of dietary magnesium with chronic pain while controlling for socially and clinically relevant confounders. However, a cross-sectional analysis can only show an association and does not confirm causality. Although we adjusted for confounding by several important factors, unmeasured confounders may be present. We also cannot rule out reverse causality to explain the association between magnesium intake and chronic pain since poor dietary intake of magnesium could be a result of chronic pain. Although the assessment of the predictor and outcome are self-report variables trained interviewers were used to collect the data and the methods were validated and consistent over the years of data collection [46]. The use of magnesium intake at only one time point may not reflect long term intake. The prevalence of pain in this dataset (15.5%) is lower than recently reported values of 20% or more [1], possibly reflecting increases in the general population over the last two decades.

## 5. Conclusions

Magnesium intake adjusted for body weight appears protective for chronic pain. The effect is larger in women than men. Dietary or supplemental magnesium intake may be a safe, low cost alternative. Future clinical trials should explore the use of magnesium as the primary treatment or as an adjunct therapy for chronic pain.

## Figures and Tables

**Figure 1 nutrients-12-02104-f001:**
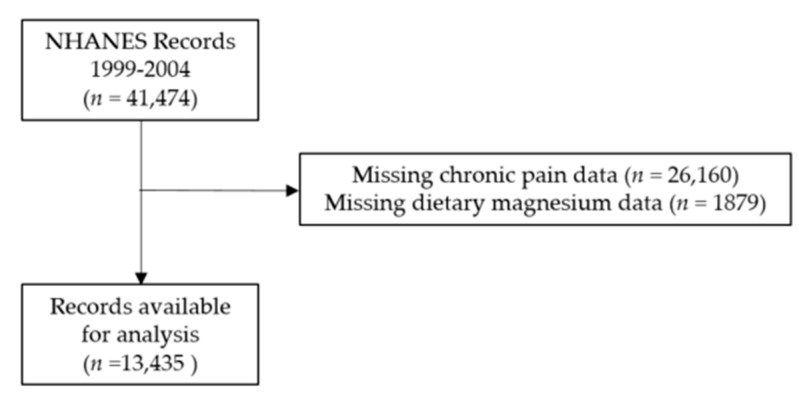
Consort diagram. National Health and Nutrition Examination Survey (NHANES)**.**

**Figure 2 nutrients-12-02104-f002:**
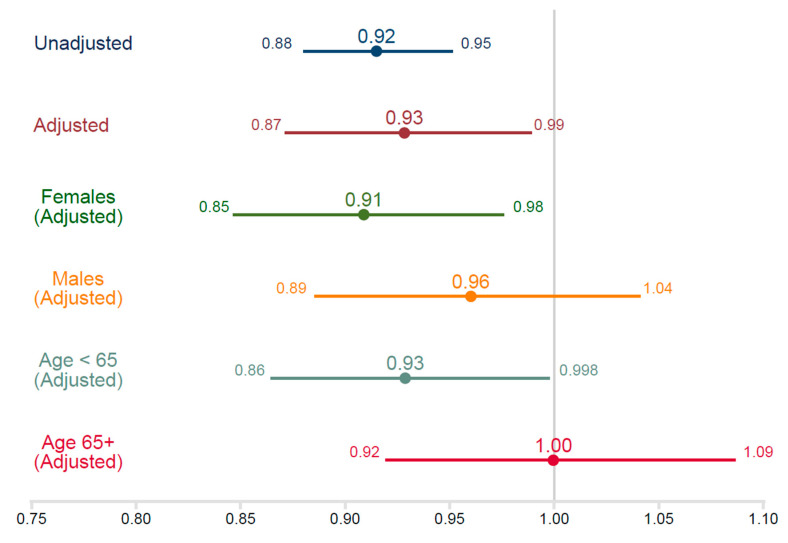
Odds ratios by logistic regression on chronic pain.

**Table 1 nutrients-12-02104-t001:** Subject Characteristics.

Variable	Control			Chronic Pain			
	N	Mean or Percent	Standard Deviation	N	Mean or Percent	Standard Deviation	*p*
Mg intake/body weight (mg/kg)	11,280	3.7	2.1	1898	3.3	1.9	<0.001
Total Mg intake (mg)	11,488	275	145	1947	262	142	0.003
Total Energy Intake (kcal/day)	11,488	2111	1015	1947	2044	1016	0.006
Mg intake/energy intake (mg/kcal)	11,486	0.14	0.05	1947	0.13	0.05	0.90
Mg intake ≥ EAR	11,488	35%		1947	32%		0.015
Age (year)	11,488	49.4	19.3	1947	52.4	17.1	<0.001
Age ≥ 65	11,488	27%		1947	26%		0.98
Male Gender	11,488	48%		1947	43%		0.004
BMI (kg/m^2^)	11,197	28.2	6.1	1869	29.6	7.1	<0.001
Race							
Non-Hispanic White	5637	49.1%		1140	58.6%		<0.001
Non-Hispanic Black	2195	19.1%		359	18.4%		
Mexican American	2714	23.6%		311	16.0%		
Other Hispanic	541	4.7%		72	3.7%		
Other Race	401	3.5%		65	3.3%		
Daily Activity							<0.001
Sitting	2807	24.5%		639	32.9%		
Walking	6163	53.7%		918	47.3%		
Walking and Light Lifting	1758	15.3%		275	14.2%		
Walking and Heavy Lifting	749	6.5%		110	5.7%		
Alcohol Intake per Week	10,889	2.9	7.1	1869	2.9	8.6	0.80
Current Smoker	11,471	20%		1946	29%		<0.001
High School Graduate	11,463	68%		1946	68%		0.011
Low Household Income	10,713	48%		1848	54%		0.003
Currently Married or Living as Married	11,081	62%		1885	62%		0.026
Kidney Dysfunction	11,151	12%		1891	16%		<0.001
Atherosclerotic Disease	11,488	10%		1947	20%		<0.001
Rheumatoid Arthritis	11,475	5%		1938	12%		<0.001
Osteoarthritis	11,475	6%		1938	17%		<0.001
Chronic Obstructive Pulmonary Disease	11,488	4%		1947	9%		<0.001
Diabetes Mellitus	11,488	12%		1947	17%		<0.001
Obese	11,197	31%		1869	41%		<0.001

Magnesium (Mg); Estimated Average Requirement (EAR).

**Table 2 nutrients-12-02104-t002:** Multivariate logistic regression analyses of magnesium intake per day per kg on chronic pain.

	All		Females		Males		Age < 65		Age 65+	
	(*n* = 11,105)		(*n* = 5780)		(*n* = 5325)		(*n* = 8188)		(*n* = 2917)	
	*OR*	95% *CI*	*OR*	95% *CI*	*OR*	95% *CI*	*OR*	95% *CI*	*OR*	95% *CI*
Mg intake/body weight (mg/kg)	0.93 *	0.87, 0.99	0.91 *	0.85, 0.98	0.96	0.89, 1.04	0.93 *	0.86, 0.998	1.00	0.92, 1.09
Age (year)	1.00	0.99, 1.00	1.00	0.99, 1.00	1.00	0.99, 1.01	--	--	--	--
Male Gender	0.79 **	0.68, 0.91	--	--	--	--	0.81 *	0.69, 0.95	0.70 *	0.52, 0.94
Race										
White	1.00	--	1.00	--	1.00	--	1.00	--	1.00	--
Mexican American	0.49 **	0.40, 0.60	0.62 **	0.47, 0.81	0.35 **	0.25, 0.50	0.45 **	0.36, 0.56	0.66	0.43, 1.01
Other Hispanic	0.70 *	0.49, 1.00	0.62	0.35, 1.09	0.79	0.43, 1.45	0.64 *	0.44, 0.94	0.79	0.39, 1.60
Non-Hispanic Black	0.76 *	0.62, 0.93	0.73 *	0.53, 0.92	0.78	0.56, 1.08	0.70 *	0.55, 0.89	0.81	0.55, 1.19
Other Race	0.88	0.58, 1.32	0.99	0.61, 1.62	0.72	0.39, 1.33	0.78	0.51, 1.19	0.97	0.43, 2.19
Low Household Income	1.20 *	1.00, 1.44	1.18	0.95, 1.45	0.72	0.39, 1.33	1.39 **	1.14, 1.69	0.82	0.61, 1.12
High School Graduate	0.93	0.80, 1.07	0.93	0.77, 1.13	0.94	0.73, 1.21	0.92	0.78, 1.09	0.79	0.55, 1.12
Currently Married	1.37 **	1.18, 1.59	0.93	0.77, 1.13	1.75 **	1.34, 1.2.28	1.38 **	1.18, 1.62	1.03	0.78, 1.35
Total Daily Calorie Intake	1.00	1.00. 1.00	1.00	1.00. 1.00	1.00	1.00. 1.00	1.00	1.00, 1.00	1.00	1.00, 1.00
Daily Activity										
Sitting	1.00	1.00. 1.00	1.00	1.00. 1.00	1.00	1.00. 1.00	1.00	1.00. 1.00	1.00	1.00. 1.00
Walking	0.86	0.74, 1.01	0.82	0.65, 1.04	0.92	0.76, 1.12	0.86	0.70, 1.04	1.00	0.76, 1.32
Walking and Light Lifting	0.88	0.74, 1.05	0.93	0.69, 1.23	0.85	0.63, 1.13	0.86	0.70, 1.06	0.98	0.57, 1.67
Walking and Heavy Lifting	1.04	0.81, 1.33	1.39	0.75, 2.59	0.95	0.75, 1.20	1.00	0.76, 1.31	1.40	0.50, 3.93
Current Smoker	1.69 **	1.43, 2.00	1.70 **	1.39, 2.08	1.74 **	1.39, 2.17	1.59 **	1.33, 1.89	1.16	0.76, 1.78
Alcohol Intake per Week	1.01	1.00, 1.01	0.99	0.97, 1.02	1.01 *	1.00, 1.02	1.01	1.00, 1.01	1.00	0.98, 1.02
Kidney Dysfunction	1.11	0.88, 1.39	1.04	0.78, 1.40	1.26	0.86, 1.83	1.40 *	1.02, 1.93	0.95	0.72, 1.26
Atherosclerotic Disease	1.56 **	1.24, 1.94	1.55 *	1.15, 2.08	1.51 *	1.14, 2.01	1.84 **	1.32, 2.57	1.70 **	1.35, 2.15
Rheumatoid Arthritis	3.27 **	2.64, 4.05	2.61 **	1.97, 3.44	4.34 **	3.09, 6.11	3.86 **	2.86, 5.21	2.03 **	1.39, 3.00
Osteoarthritis	3.04 **	2.39, 3.87	3.02 **	2.27, 4.01	3.08 **	2.14, 4.43	3.98 **	2.98, 5.31	2.05 **	1.46, 2.86
Chronic Obstructive Pulmonary Disease	1.70 **	1.32, 2.20	1.68 **	1.28, 2.21	1.78 *	1.10, 2.81	1.76 *	1.21, 2.57	1.58	0.98, 2.56
Diabetes Mellitus	1.30 *	1.08, 1.58	1.48 **	1.20, 1.82	1.10	0.82, 1.47	1.29 *	1.01, 1.65	1.39 *	1.01, 1.03

* *p* < 0.05; ** *p* ≤ 0.001; Magnesium (Mg); independent odds ratio when controlling for all the other variables in the model (*OR)*; confidence interval (*CI*).

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
