# Peer review of "Relationship between Magnesium Intake and Chronic Pain in U.S. Adults"

_nutrients, 2020, doi:10.3390/nu12072104_

Round 1
Reviewer 1 Report
The authors analysed the associations of magnesium intake with low back pain using the data from NHANES. Both intake data (single 24-hour recall) and the diagnosis of low back pain are self-reported. Information on the validity of both variables need to be provided in the manuscript. Even though, the potentially low quality of data need to be addressed in the discussion as limitation.
In the statistical analyses, the authors included a range of potential confounders but most of important ones were dichotomous variables. Including these confounding variables with more categories will limit residual confounding. In addition, key confounding variables including occupation and physical activity have not been taken into account.
The so-called "secondary analyses" gave completely null results, which in my opinion should be the main results. At least these should be given equal weight of importance as the results from other analyses. If necessary, the authors could investigate and discuss why different analytical approach gave different results.
Some of the statements are wrong. For example, line 127-128, how come the association with OR 0.99 is stronger than the association with OR 0.96? English editing is needed for lines 125, 129, 130, and 131. Table 1, given this is not a case-control study, there should not be a control group. Table 2, the number of participants in the younger and older groups must be wrong? since I can't imagine there were exactly the same number of participants in both groups.
I wonder why the Table 2 need to present the associations of all covariates with low back pain. They are irrelevant for the research objective. The OR of total energy intake is 1.00 (1.00-1.00). If this is the case, why was this variable included in the model?
In the first paragraph of the discussion, the summary of finding used univariate OR, this is not appropriate.
In the discussion, the authors mentioned the appropriateness of dietary change for preventing low back pain. But in this paper, the authors did not provide any information about the major resources of magnesium.
Given the cross-sectional study design and observational nature of the study, it is not appropriate to use terms such as "effect" to assume a causal association.
Author Response
Thank you for your thorough review of our manuscript. We value your feedback and hope the revisions in the updated manuscript are adequate.
1. The limitations were updated to reflect the use of self-reported data.
2. Most of the potential confounders (gender, race, medical diagnoses, etc.) were collected as binary variables. We collapsed a few others (such as marital status, income and education) because finer classifications resulted in little change in the OR of Mg on pain but introduced complex and confusing modeling terms that limit communication to the intended audience. We agree that physical activity is an important potential confounder and added it to the analysis. It did not change the results very much, but they were updates as appropriate. The physical activity variable we utilized included physical activity as a component of one's occupation.
3. The two analyses are quite different – they ask different questions and have different interpretations. As prior research has shown (Foulkes, 1997; Nielsen & Johnson, 2017), subjects of different weight require different amounts of magnesium to maintain homeostasis. Our primary hypothesis was about the intake of magnesium per kilogram of body mass, not total magnesium intake, and demonstrates an association. The secondary analysis is about total magnesium intake per day and shows no association. This suggests that magnesium intake needs to be conceptualized (and perhaps managed) quite differently in adults of different sizes.
Foulkes, R. G. (1997). Dietary reference intakes-Calcium, phosphorus, magnesium, vitamin D, and fluoride.
Nielsen, F. H., & Johnson, L. A. K. (2017). Data from controlled metabolic ward studies provide guidance for the determination of status indicators and dietary requirements for magnesium. Biological trace element research, 177(1), 43-52.
4. The results reported in line 127 were corrected in the results section and Table 2. The number of subjects by age was corrected in Table 2.
5. We agree that the main hypothesis can be expressed as the OR of magnesium intake per kg of body weight on chronic pain. It is our experience that reviewers and consumers of multivariate analyses are quite dissatisfied if the covariates are not presented. If the editor prefers, Table 2 can be made an appendix.
6. Line 144 was changed to reflect the findings of the multivariate analysis.
7. This information was added to line 171.
8. This has been corrected.
Reviewer 2 Report
Interesting paper. Even though the effects of Mg on pain are modest it merits publication.
What is the main question addressed by the research? Is it relevant and interesting? There is growing interest in the effects of magnesium. In various chronic diseases magnesium deficiency is aggravating the problems. Therefore, investigating if there is a connection of magnesium intake and chronic pain is relevant. How original is the topic? What does it add to the subject area compared with other published material? Up to now this topic has not been addressed in a large epidemiological study. Is the paper well written? Is the text clear and easy to read? Yes Are the conclusions consistent with the evidence and arguments presented? Do they address the main question posed? Yes
Author Response
Thank you for your thorough review of our manuscript. We hope our submitted revisions are acceptable.
Reviewer 3 Report
In the work the relationship between magnesium intake and chronic pain in U.S. adults were examined. The topic is very important from the healthcare point of view.
In the work there are minor spelling mistakes at (e.g. page 3 line 121 choric instead of chronic).
In Table 2 (Multivariate logistic regression analyses of magnesium intake per day per kg on chronic pain) please provide p values, at least if the impact of confounder was significant.
In the description of results, please check the odds ratios and p values ( page 3, lines 126-128) with table 2 and figure 2, especially regarding to sex (males) and age.
Author Response
Thank you for your thorough review of our manuscript. We hope the revisions in this updated manuscript are acceptable and address your concerns.
- The manuscript was reviewed and spelling errors corrected.
- Table 2 was update to reflect significant confounders
- Values on the indicated lines were checked and updated as appropriate.